# Current Molecular Markers of Melanoma and Treatment Targets

**DOI:** 10.3390/ijms21103535

**Published:** 2020-05-16

**Authors:** Kevin Yang, Allen S.W. Oak, Radomir M. Slominski, Anna A. Brożyna, Andrzej T. Slominski

**Affiliations:** 1Department of Dermatology, University of Alabama at Birmingham, Birmingham, AL 35294, USA; kyang22@uab.edu (K.Y.); siwonoak@uabmc.edu (A.S.O.); 2Division of Clinical Immunology and Rheumatology, Department of Medicine, University of Alabama at Birmingham, Birmingham, AL 35294, USA; rslominski@uabmc.edu; 3Department of Human Biology, Institute of Biology, Faculty of Biological and Veterinary Sciences, Nicolaus Copernicus University, 87-100 Toruń, Poland; anna.brozyna@umk.pl; 4Comprehensive Cancer Center, Cancer Chemoprevention Program, University of Alabama at Birmingham, Birmingham, AL 35294, USA; 5Veteran Administration Medical Center, Birmingham, AL 35294, USA

**Keywords:** melanoma, molecular pathology, diagnosis, therapy, molecular testing, genetic mutations, UV irradiation

## Abstract

Melanoma is a deadly skin cancer that becomes especially difficult to treat after it metastasizes. Timely identification of melanoma is critical for effective therapy, but histopathologic diagnosis can frequently pose a significant challenge to this goal. Therefore, auxiliary diagnostic tools are imperative to facilitating prompt recognition of malignant lesions. Melanoma develops as result of a number of genetic mutations, with UV radiation often acting as a mutagenic risk factor. Novel methods of genetic testing have improved detection of these molecular alterations, which subsequently revealed important information for diagnosis and prognosis. Rapid detection of genetic alterations is also significant for choosing appropriate treatment and developing targeted therapies for melanoma. This review will delve into the understanding of various mutations and the implications they may pose for clinical decision making.

## 1. Introduction

Melanoma represents the most lethal type of skin cancer, with an estimated 7000 deaths in 2019 in the United States [1]. Furthermore, its incidence has steadily risen since the 1960s, with approximately 96,000 new cases in 2019 [2]. Ultraviolet radiation (UVR) represents a major contributor to cutaneous melanomagenesis through its deleterious effects on the skin and direct damage to DNA [3]. These processes trigger the acceleration of tumorigenesis and thus have facilitated the emergence of malignant melanoma as a significant public health problem.

The financial burden of treating melanoma remains cumbersome. In the United States, the estimated annual cost per patient for Stage I melanoma is $2169–$14,499, while the cost for Stage IV melanoma is $34,103–$152,244 [4]. While the advent of new immunotherapies has helped to reduce the mortality rate over the last decade, melanoma remains difficult to treat. Given the expensive costs for treatment of advanced-stage melanoma, diagnosing and treating melanoma at an early stage is crucial. However, the heterogeneous morphological and histopathological appearance of these tumors can make diagnosis challenging.

As per the 2018 WHO classification, melanoma was newly divided into three classes: melanomas associated with cumulative solar damage (CSD), melanomas not associated with CSD, and nodular melanoma [5]. Pathways of melanoma associated with CSD include superficial spreading, lentigo maligna, and desmoplastic melanomas. Superficial spreading melanoma is the most common subtype and is noted for its early radial growth, followed by vertical growth and subsequent invasion into the dermis [6]. Melanomas not associated with CSD are subclassified into spitzoid, acral, mucosal, and uveal melanomas and melanomas arising in congenital and blue nevi. Finally, nodular melanoma is distinctive for its early progression to vertical growth [6,7]. This new classification better delineates the mutagenic changes that arise in melanoma formation.

In addition to the formulation of melanoma on the skin, uveal melanoma has emerged due to its morbidity and mortality. Uveal melanoma originates from melanocytes that are located in the iris, choroid, or ciliary body [8,9]. Uveal melanoma has an estimated annual incidence of 5.1 cases per million and comprises approximately 3%–5% of all melanomas [8,9,10]. Although relatively rare, this malignancy is highly lethal due to its rapidly metastatic nature. Interestingly, uveal melanoma may be associated with similar risk factors to those of cutaneous melanoma, including fair skin and sunlight exposure [10,11,12]. Treatment is limited but has revolved around radiation and possible enucleation [13].

Diagnostic and therapeutic molecular markers have been increasingly used to assist in histopathological assessment of these tumors. These markers are not only helpful for diagnosing melanoma, but also in distinguishing certain subtypes that may otherwise be difficult to identify (Table 1). In addition, therapeutic markers can guide the selection of treatment with the development of novel targeted therapies (Table 2).

## 2. Molecular Pathways of Melanoma Formation

Cutaneous melanomagenesis can generally be traced to mutations in signaling pathways critical to cell survival. Most notably, the mitogen-activated protein kinase (MAPK) pathway regulates cell growth, proliferation, differentiation, and apoptosis [77,78,79,80]. Mutations along this pathway result in overamplification of signaling, leading to cell cycle dysregulation and uninhibited cell growth. The MAPK pathway is activated by binding of a growth factor to a receptor tyrosine kinase (RTK) on the surface of the cell and stimulates the GTPase activity of RAS. The signal is propagated down through the cascade of RAF, MEK, and ERK, which enters the nucleus to activate transcription factors promoting the cell cycle [81,82,83,84,85,86]. The PI3K/AKT pathway regulates cell growth and proliferation [87,88,89,90]. PI3Ks can be similarly activated by binding of RTK to a growth factor or directly stimulated by RAS [91,92]. PI3K subsequently phosphorylates its substrate PIP_2_ on the cell membrane into PIP_3_, helping to recruit and activate AKT [93,94,95]. AKT promotes cell growth and survival through multiple effectors, such as mTOR, Bad, and Mdm2 [87,88,89]. Mutations to these regulatory signals such as the oncogene *NRAS* or the tumor suppressor *PTEN* can occur alone or even in addition to other mutations in melanoma [91,96].

In contrast to cutaneous melanoma, uveal melanoma tends to develop from different mutations along the MAPK or PI3K/AKT pathways. The most common mutations are in *GNAQ* or *GNA11*, which can lead to overactivation of both pathways. These genes are responsible for encoding the Gα subunit of G proteins, leading to a constitutively active GTP-bound state [70,71,72]. *GNAQ* and *GNA11* mutations may also increase activity through the Hippo pathway. The Hippo pathway has been identified for its role in cell homeostasis and mammalian organ size, including heart, liver, and pancreas [97,98,99,100]. *GNAQ* and *GNA11* mutations result in downstream activation of YAP/TAZ to stimulate melanomagenesis [101]. Uveal melanoma has been thought to result from an initiating *GNAQ/GNA11* mutation, followed by a secondary BSE event from mutations in the genes *BAP1*, *SF3B1*, and *EIF1AX* [13,102].

## 3. Molecular Markers

Melanoma is generally diagnosed by assessment of skin histological and architectural features but can be prone to subjectivity. Further, traditional characteristics of melanomas such as thickness or mitotic rate can be inaccurate in diagnosis and prognosis. For these reasons, there is an ever-present search for novel detection methods. Detecting molecular markers or genetic alterations has emerged as an innovative form of testing that guides therapeutic decisions and aids the diagnosis of histologically challenging cases. Sequencing studies have illuminated the role of UV exposure in different mutations that lead to melanoma. For example, identifying UV signature mutations, such as C → T and CC → TT substitutions, can provide an idea of the underlying impact of UV radiation [103]. Whole-genome sequencing has revealed the different mutations that contribute to the development of UV-dependent and -independent melanomas [104]. Methods including comparative genomic hybridization (CGH), fluorescence in situ hybridization (FISH), and quantitative gene expression profiling contribute to the detection of genetic mutations and determination of expression levels. Tests in clinical use include DecisionDx-Melanoma (Castle Biosciences), myPath Melanoma (Myriad Genetics), and Pigmented Lesion Assay (DermTech, Inc.), which profile a wide array of genes [105,106,107,108]. As these tests become more refined, the meaning of various markers in diagnosis and therapy of melanoma has expanded as well. These markers can be represented by melanoma mutations, gene polymorphisms, signaling receptors, and melanin pigment. For this review, we will discuss the significance of these in the context of their role in prognostic and diagnostic value, the melanin synthesis pathway, and targeted therapeutics.

### 3.1. Prognostic or Diagnostic Markers

#### 3.1.1. GNAQ/GNA11

*GNAQ* and *GNA11* mutations result in overamplification of signaling through the MAPK and PI3K pathways via blocking GTPase activity. G proteins become active when bound to GTP and are inactivated by GTPase hydrolysis to GDP. With *GNAQ* and *GNA11* mutations, GTP is persistently bound to the G protein and lead to constitutive downstream signaling [9,70,71,72]. These mutations are mutually exclusive and are detected in approximately 80%–90% of cases of uveal melanoma [29,109]. However, they are known to occur with *BAP1* and *SF3B1* mutations, with *GNAQ*/*GNA11* mutation representing the initial event [13,102]. Because uveal melanoma rapidly metastasizes, in such advanced stage cases, identification of the primary tumor can be difficult. Analysis of oncogene status showing positive *GNAQ* or *GNA11* expression can be a valuable diagnostic tool to differentiate uveal melanoma from other types of melanoma and cancers [109]. While *GNAQ* and *GNA11* mutations can also be found in cutaneous melanoma, these cases are extremely rare [110].

The evidence for the prognostic value of *GNAQ* and *GNA11* mutations is limited. Multiple studies have shown that the presence of *GNAQ* or *GNA11* mutations is not associated with metastatic progression or patient outcomes [111,112]. In addition, no difference has been found in survival between patients harboring the *GNAQ* mutation versus patients with the *GNA11* mutation [29,111].

#### 3.1.2. CDKN2A

Mutations in the *CDKN2A* gene are the most common alteration in hereditary melanoma, with presence in 40% of families with strong family history [20,21,113]. This gene encodes the p16 protein, which inhibits cyclin-dependent kinase (CDK) 4 and 6, and the p14^ARF^ protein. Mutations in *CDKN2A* thus result in hyperphosphorylation of retinoblastoma protein (RB1), releasing the E2F1 transcription factor to promote cell cycle progression from G1 to S. In addition, loss of p14^ARF^ function promotes the ubiquitination of p53, subsequently reducing cell cycle arrest and apoptosis [48,49,50,114].

Those with the *CDKN2A* mutation have been shown to develop multiple melanomas and significantly more dysplastic nevi, including presentations consistent with dysplastic nevus syndrome [22]. Interestingly, one study found *CDKN2A* penetrance varied with geographic location, postulating a correlation with UV exposure as highest penetrance by age 80 in families from Australia [20,115]. Histological analysis of *CDKN2A*-mutated familial melanomas revealed a greater association with the superficial spreading subtype as compared with *CDKN2A*-wild type familial melanomas [116]. Furthermore, in a Swedish study, familial melanoma cases with the *CDKN2A* mutation were associated with a younger age at onset and worse survival than those without the mutation. That study suggested that dysregulation of the cell cycle with *CDKN2A* mutations may exacerbate mutational load and increase tumor aggression [117]. On the other hand, an Italian retrospective cohort study found no association of the mutation with worse survival [118].

#### 3.1.3. BAP1

*BAP1* is a tumor suppressor gene with a poorly understood mechanism in melanoma development but has been implicated as a deubiquitinase of cell cycle genes [73,119]. *BAP1* mutations are associated with monosomy 3, which is associated with metastatic uveal melanoma [13,33,120]. Multiple other studies since have confirmed the correlation of *BAP1* mutations with metastasis, tumor aggression, and worse prognosis in uveal melanoma [30,121,122]. Notably, *BAP1* was found to be mutated in early tumorigenesis and not with progression to metastasis [123].

Germline mutations of *BAP1* have also been identified, suggesting a hereditary form of uveal melanoma. In fact, *BAP1* is the most common mutation found in familial uveal melanoma, with an estimated frequency of 8%–50% of such cases [31]. Genetic testing is helpful on a case-by-case basis, particularly with suggestive family history. Similar to somatic mutations, germline mutation of *BAP1* was highly associated with metastasis as compared with uveal melanoma without *BAP1* mutation [124]. In another study, somatic mutations were found to have a greater risk of metastasis as compared to germline mutations. As a result, determining *BAP1* status in cases of uveal melanoma can be useful to understanding the risk of metastasis [125].

The *BAP1* tumor predisposition syndrome caused by germline *BAP1* mutations is not only associated with cutaneous melanoma [126,127]. Rarely, somatic mutations may also lead to cutaneous melanoma [126]. Kumar et al. suggested that *BAP1* may have differential roles in uveal and cutaneous melanoma cells [128]. In contrast to its role as a tumor suppressor gene, *BAP1* expression in cutaneous melanoma was found to promote growth and survival of cells [128]. *BAP1*-inactivated nevus (BAPoma) is a relatively new entity of an atypical spitzoid tumor, and can serve as an early sign of BAP1 tumor predisposition syndrome [32,129]. The conflicting roles of *BAP1* underscore the dearth of understanding into its mechanism. Therapeutic targeting of BAP1 has focused on its role in DNA double-strand break repair via homologous recombination [130]. Mutations in *BAP1* thus have been hypothesized to rely on alternative mechanisms of DNA repair with poly (ADP-ribose) polymerase (PARP) emerging as a target for its role in base-excision and nucleotide excision repair [131]. A phase II trial is recruiting to evaluate the effect of niraparib, a PARP inhibitor, in the treatment of uveal melanoma (NCT03207347) [74].

#### 3.1.4. SF3B1

*SF3B1* encodes a subunit of splicing factor 3b, and mutations therefore result in aberrant splicing of pre-mRNA into mature mRNA [34,75]. *SF3B1* mutations are characterized by disomy 3 and noted to be a marker of good prognosis for uveal melanoma and found in younger patients [13,33,120,122,132]. While tumors bearing the mutation often metastasize, this can take many years and they are thus thought to have intermediate risk for metastasis [35]. Metastasis is thought to occur with the development of additional oncogenic mutations [123].

In one study sequencing melanoma samples, the *SF3B1* R625 mutation was found in two out of 231 cutaneous melanoma samples. That report also noted those cases to be metastatic and, similar to *BAP1*, posed the question of differential roles of *SF3B1* depending on the type of melanoma cells [133].

#### 3.1.5. EIF1AX

*EIF1AX* is important in regulating protein translation, as it encodes for a eukaryotic initiation factor that serves to stabilize the ribosome [34,76]. These mutations are also characterized by disomy 3 [33]. Cases of uveal melanoma with positive *EIF1AX* mutations rarely metastasize and other genetic alterations are thought to occur when metastasis does occur [35,37,122,123]. It follows that these cases are also generally associated with good prognosis [122].

#### 3.1.6. VDR

The vitamin D receptor (VDR), after binding the active form of vitamin D, 1,25-dihydroxyvitamin D_3_ (1,25-(OH)_2_D_3_), complexes with retinoid X receptor (RXR) to form a trimolecular complex that translocates to the nucleus and binds to VDR response elements (VDRE) on DNA to function as a transcription factor [52,53,54,134]. Further, 1,25(OH)_2_D_3_ has been shown to protect against melanoma by inhibiting proliferation, regulating growth factor activity, and promoting apoptosis [135,136,137,138,139,140,141]. Loss of this activity is thought to contribute to the formation and progression of melanoma. In an early study of *VDR* in melanoma patients, expression was found to be inversely correlated with progression from normal skin to melanocytic nevus to melanoma, suggesting a potential role in aiding differentiation between nevi and early melanoma [23,24]. Reduction in *VDR* expression was also associated with tumor progression, higher mitotic rates, and shorter survival time [23,24] (Figure 1). Localization of *VDR* expression is also important as lower expression of cytoplasmic *VDR* was found more commonly in melanoma as compared to nevi and also associated with tumor size [142]. However, nuclear *VDR* expression in nevi versus melanoma contrasts with this [23,142].

Various polymorphisms of *VDR* have also been found to affect the risk and prognosis of melanoma, but a consistent pattern has not been identified [141]. Orlow et al. first studied 38 common *VDR* single-nucleotide polymorphisms (SNPs) and discovered six of these to be associated with increased risk of melanoma development and two with decreased risk [145]. In a later study by the same group, eight SNPs were found to be associated with improved survival in melanoma [146]. Furthermore, the impact of various SNPs on melanoma survival also depends on the amount of sun exposure around time of melanoma diagnosis [147]. However, a 2009 study analyzing six *VDR* SNPs found no significant change in outcomes with the exception of worse outcome in patients with the BsmI polymorphism and low vitamin D levels [148]. Overall, the understanding of *VDR* variants in melanoma remains poor [149].

Recently, *VDR* has also been proven to exist in uveal melanocytes and melanomas as well. While *VDR* expression was not found to be associated with histopathological characteristics of uveal melanoma, an inverse correlation was shown between expression level and the degree of tumor pigmentation [150]. In light of the positive correlation between uveal tumor pigmentation and metastatic risk, the pattern of *VDR* expression in uveal melanoma matches closely with that in cutaneous melanoma, implicating similar utility in diagnosis.

*VDR* expression has been shown to be negatively correlated with overall prognosis in melanoma patients, painting an important picture for vitamin D-based therapy [23,24,140]. In 1981, Colston et al. were the first to discover that 1,25(OH)_2_D_3_ inhibits the proliferation of human melanoma cells [151]. Since then, similar results have been elicited by 1,25(OH)_2_D_3_ in other human melanoma cell lines and animal models [141,152,153,154,155]. Importantly, VDR mediates the effects of 1,25(OH)_2_D_3_ [156]. A recent study by Wasiewicz et al. further confirmed this, as resistance to the anti-proliferative properties of vitamin D was seen in the human melanoma cell line SK-MEL-188b, a *VDR*^-/-^ subline [157]. In fact, metabolites and analogs of 1,25(OH)_2_D_3_ are able to exert similar effects through VDR [152]. This becomes especially important because 1,25(OH)_2_D_3_ is limited in its therapeutic use due to its hypercalcemic effects. Analogs with modified side chains, such as calcipotriol, have been demonstrated to inhibit proliferation with low calcemic activity [158,159,160]. Similar effects have been demonstrated for noncalcemic vitamin D_3_ hydroxyderivatives [159,160,161], which are products of the CYP11A1 action of the side chain of vitamin D_3_ [63,162]. CYP11A1, in addition to adrenals, placenta and sex organs [163], is also expressed in several peripheral tissue including immune system and skin [164,165,166,167,168,169,170,171]. The anti-melanoma effect of 20-hydroxyvitamin D_3_ (20(OH)D_3_) was also reported in an in vivo model of melanoma [153]. It must also be noted that while there was an inverse correlation between CYP27B1 (enzyme activating vitamin D_3_) and disease progression [143], an inactivating enzyme, CYP24A1, also showed an unexpectedly inverse correlation [144] (Figure 1). This finding was possibly due to alternative hydroxylation pathways forming other anti-tumorigenic dihydroxy-derivatives [144].

In the clinical setting, vitamin D intake has not been associated with melanoma risk; however, its role as adjuvant therapy in melanoma is unknown [172]. A phase 2 trial (Mel-D) is underway in Australia to assess the safety and progression-free survival results of high dose vitamin D therapy in patients who have had surgical excision of melanoma and are at high risk for recurrence [55]. In a Belgian phase 3 trial (ViDMe), relapse-free survival is being assessed in patients treated with cholecalciferol after primary excision of melanoma [56]. The results of these trials will provide key understanding for the therapeutic utility of vitamin D in combination with the assessment of *VDR* expression.

#### 3.1.7. MC1R

The melanocortin 1 receptor (*MC1R*) has been identified as a major gene in developing sporadic melanoma [25,57,173]. MC1R responds to melanocyte-stimulating hormone (MSH) to regulate melanogenesis and skin pigmentation. Variants in the *MC1R* gene have been shown to directly and indirectly induce melanomagenesis. In melanocytes, MC1R activates DNA repair and reduces oxidative stress. Thus, *MC1R* polymorphisms can exacerbate the UV-induced DNA damage and promote tumor formation [174,175,176,177]. In addition, variants of *MC1R* upregulate pheomelanin production, which is characterized by a phenotype of fair skin and red hair and susceptibility to UV light [178].

Multiple studies have demonstrated that *MC1R* variation confers increased risk for melanoma [26,179,180]. While this association was thought to be driven by the predisposition to fair skin, multiple large studies have shown that the presence of a *MC1R* variant was associated with development of melanoma, independent of all other risk factors including skin type [25,181]. However, in a study stratified by sex, such an association was only discovered in females [181]. While the presence of *MC1R* variation has not been associated with histopathologic characteristics, it was found to correlate with tumor presentation on the arms, which may provide additional support for its UV-risk independence [182]. Melanomas associated with germline mutations of *MC1R* have also been shown to have a significantly higher somatic mutational burden, suggesting a higher susceptibility to tumorigenesis in these patients [183]. Notably, *MC1R* variants have been shown to increase the penetrance of *CDKN2A* mutations, doubling the risk for melanoma [184].

#### 3.1.8. MITF

Microphthalmia transcription factor (*MITF*) regulates melanocyte development, differentiation, and function [58,59]. *MITF* has been shown to serve as a sensitive and specific marker for distinguishing melanoma from histologically similar nonmelanocytic tumors [185,186,187]. *MITF* has also been found to regulate phenotype switching, wherein low expression leads to increased invasiveness of melanoma cells and high expression leads to decreased invasiveness [188]. Indeed, Cheli et al. showed that *MITF* silencing in mouse and human melanoma cells enhanced tumorigenicity and metastasis [189]. Investigation of the clinical significance of *MITF* expression in melanoma samples showed its positive correlation with survival. Higher expression was also associated with negative lymph node status [27].

In addition to its utility as a diagnostic marker for melanoma, *MITF* has also been found to increase susceptibility to the co-occurrence of melanoma and renal cell carcinoma (RCC). The presence of the Mi-E318K germline mutation in *MITF* was associated with a fivefold increase in the risk of developing melanoma as compared to patients without the mutation [28,190]. Recently, Ciccarese et al. also reported the association of this variant nodular with the development of nodular melanoma and dysplastic nevi [191]. In cases of familial melanoma, testing for the *MITF* mutation may be helpful.

#### 3.1.9. HAPLN1

Age represents another risk factor predicting inferior survival in melanoma patients [192]. One explanation is that age-related changes degrade the extracellular matrix (ECM) in the skin and thus promote the growth and migration of melanoma [65]. With age, fibroblasts secrete fewer ECM components with hyaluronan and proteoglycan link protein 1 (HAPLN1) identified as major components [65]. Subsequent study of HAPLN1 revealed its utility as a potential prognostic biomarker. High expression of HAPLN1 in lymphatic tissue was associated with a 56% decrease in death [193]. Thus, determining HAPLN1 expression levels may be particularly helpful in elderly patients.

### 3.2. Members of the Melanin Synthesis Pathway

#### 3.2.1. Melanin

The synthesis of melanin pigment is characteristic of melanocytes and plays various roles in those cells, including as a free radical scavenger to protect against UV radiation [60,61,62,194,195]. Melanin also serves as a differentiation marker of normal and cancerous melanocytes and thus serves as a suitable marker to distinguish melanoma from other tumors [196].

The degree of melanization has also been shown to impact the aggressiveness and prognosis of melanoma. In one study, patients with stage 3 or 4 melanotic melanoma had poorer survival as compared to patients with amelanotic or hypomelanotic melanomas [197]. Such an association may in part be attributed to the upregulation of *HIF-1α* by melanogenesis and subsequent downstream stimulation of angiogenesis and cellular metabolism, promoting tumor aggression [198]. Similarly, correlation of melanogenesis to metastasis and death has been shown in uveal melanoma [199,200,201]. On the flip side, a large population-based study showed a twofold increased risk of death for amelanotic compared to pigmented melanoma [202]. To explain this, Sarna et al. have suggested that melanin mechanically reduces the elasticity of melanoma cells and thus inhibits metastasis [203,204].

The anti-oxidative properties of melanin represent an adaptive function of melanocytes to protect the skin from UV radiation. However, the effects of radio-, photo-, or chemotherapy can be unintentionally blunted [205]. One of the early studies investigating this phenomenon by Brozyna et al. showed that inhibition of melanogenesis by N-phenylthiourea (PTU) or D-penicillamine in human melanoma cells increased the sensitivity to killing by gamma rays [206]. Soon after, melanogenesis inhibition was also shown to amplify the cytotoxicity of cyclophosphamide chemotherapy in human melanoma cells [207]. Of note, intermediates of melanogenesis show immunosuppressive effects [207,208,209,210]. In a retrospective melanoma cohort study in Poland, among those treated with radiotherapy, mean survival after therapy and overall survival were more than twofold higher in amelanotic melanomas as compared to pigmented melanomas [205]. Although the diagnosis of an amelanotic melanoma is typically associated with a poorer prognosis, this may be due to its atypical clinical morphology that hinders its diagnosis. This is evidenced by amelanotic melanoma’s higher American Joint Committee on Cancer tumor stage at the time of diagnosis when compared to that of pigmented melanoma [202]. Interestingly, inhibition of melanogenesis also potentiates the efficacy of vitamin D therapy [161]. *VDR* expression has been shown to be lower in pigmented melanoma cells compared with nonpigmented cells [161]. Accordingly, nonpigmented cells were more sensitive to vitamin D treatment than were pigmented cells [158,161]. Overall, the potential role for melanogenesis inhibition as adjuvant treatment to primary therapy of melanoma has been demonstrated with clinical application standing as the next step.

#### 3.2.2. Melanogenesis Related Proteins (TYR, TRP1, TRP2)

Melanin is synthesized in melanocytes from L-tyrosine through a series of enzymatic reactions leading to production of variety of intermediates of melanogenesis that are biologically active [61,63,64]. Among the proteins involved are tyrosinase (TYR) and tyrosinase-related proteins 1 and 2 (TRP1, TRP2). TYR has become commonplace in the diagnosis of melanoma for its high sensitivity and specificity [211,212,213,214]. Recently, TYR expression was found to correlate with melanocyte differentiation and thus may serve to distinguish between melanoma and benign nevi [215]. TRP1 and TRP2 have also been identified as differentiation markers from the class of melanogenesis-related proteins [216,217]. TRP1 mRNA expression has been correlated with worse survival and depth of invasion; however, TRP1 protein expression was found to inversely correlate with tumor stage and also had no association with patient survival [218,219].

### 3.3. Therapeutic Targets

#### 3.3.1. B-raf

B-raf is one of the signaling kinases down the MAPK pathway. *BRAF* mutations comprise the most common genetic alteration in cutaneous melanoma with its presence ranging from 40% to 60% of cases [14,15,220]. Mutations in this oncogene lead to constitutive activation of the MAPK pathway. The most common *BRAF* mutation is V600E, which represents 80% of alterations in the gene [14]. The V600K and V600R mutations are other known *BRAF* mutations [14,220]. Studies have shown that V600E expression is associated with the superficial spreading subtype, younger patient age, and skin sites without chronic sun-induced damage, such as the extremities [16,17,221]. In contrast, V600K mutations are correlated with skin sites with CSD, such as the head and neck, and patients of older age [222]. These associations underscore the incompletely understood role of UV radiation in the development of mutations leading to melanoma.

Recently, whole-genome sequencing of benign melanocytic nevi showed the presence of *BRAF* mutations, in addition to *NRAS* mutations, with mutational load positively correlated with UV exposure; lower mutational loads were observed in congenital nevi [223]. Similar observations were found in dysplastic nevi with high mutational load as a key distinction between the benign tumors and melanoma. *BRAF* mutations were thought to be independent of UV exposure due to the absence of UV signature mutations but the consideration of “noninformative” mutations has shifted that belief [41,224]. Of note, Bauer et al. showed that *BRAF* mutations could not be detected in congenital melanocytic nevi, further suggesting the role of moderate UV exposure in introducing such mutations in the skin [225]. Accordingly, *BRAF* mutations are associated with melanomas from anatomic locations with intermittent sun exposure, such as the trunk and extremities [226]. Better understanding of the development of these mutations can help to track the rare transformation of benign nevi to malignant melanoma and distinguish the two when histology is equivocal [227]. Refined sequencing technology is also critical for determining the mutational load, which may help evaluate tumor malignancy.

Previously, it had been thought that the presence of *BRAF* mutations were not associated with worsening prognosis or tumor proliferation [220,228]. Abd Elmageed et al. showed that cellular localization of the *BRAF* mutation to either the nucleus or cytoplasm was associated with different clinicopathological features. Positive expression of V600E in the nucleus as compared to the cytoplasm was correlated with worse tumor stage, lymph node metastasis, and depth of invasion [229]. Vemurafenib, dabrafenib, and encorafenib are FDA-approved drugs that have been developed as inhibitors of *BRAF* V600 mutations [38,39,40]. While monotherapy with *BRAF* inhibitors shows good efficacy against *BRAF*-mutant melanomas, patients can easily develop resistance through upregulation of RTKs or *NRAS* [230,231]. As a result, combination therapy with *BRAF* and *MEK* inhibitors, including trametinib and cobimetinib, has become standard due to multi-pronged blockade of melanoma growth pathways. The COMBI-v and COMBI-d phase 3 trials showed greater than twofold increases in survival when comparing dabrafenib and trametinib combination to vemurabfenib or dabrafenib monotherapy [232,233]. Newer combinations of encorafenib and binimetinib potentially show even greater efficacy in *BRAF*-mutant melanomas [38]. Maintaining long-term response is difficult, with just 20% of patients in one study remaining progression free [234]. Ongoing research is investigating the addition of CDK, MAPK, or immune checkpoint inhibitors to combat resistance [231,235,236]. Recently, Misek et al. identified RhoA GTPases as a potential pathway for *BRAF* resistance and that inhibition of the pathway by Rho kinase (ROCK) inhibitors promotes resensitization to *BRAF*-targeting therapy [237]. Because of the success of target therapies, identifying patients with *BRAF*-mutant melanomas is paramount. Vemurafenib had been associated with a paradoxical upregulation of the MAPK signaling pathway in cells with wild-type *BRAF* [238]. In turn, this paradoxical upregulation had been linked to increased incidence in cutaneous SCCs that harbor *RAS* mutation, as well as reported cases of dysplastic nevi and wild-type *BRAF* melanomas, in patients after vemurafenib treatment [239,240].

#### 3.3.2. N-ras

The N-ras GTPase is critical in the transduction of extracellular growth signals via both the MAPK and PI3K/AKT pathways. When the oncogene is mutated, GTPase activity is reduced, resulting in a constitutively active GTP-bound G protein to propagate downstream signals [15,41,85,91,241,242]. Generally, *NRAS* mutations occur independently of *BRAF* mutations but dual expression has been reported [243]. In contrast to *BRAF* mutations, *NRAS*-mutant melanomas are correlated with the nodular subtype and found in CSD skin [18]. The association of *NRAS* and skin with CSD suggests that such mutagenesis is induced by UV irradiation [244]. Positive *NRAS* expression was also associated with a lower grade of tumor-infiltrating lymphocytes and a higher tumor stage [245]. The prognostic value of identification of *NRAS* mutation is unclear. A large cohort study from the M.D. Anderson Cancer Center revealed shorter survival from metastatic melanoma [17]. Others have reported, however, that overall survival is not different between WT and *NRAS*-mutated melanomas [246].

Along with *BRAF* mutations, *NRAS* mutations are among mutations found in melanocytic and dysplastic nevi and melanomas, with high mutational load as a notable discriminant favoring the latter in diagnosis [223,227]. Notably, *NRAS* mutations were also commonly found in congenital melanocytic nevi, suggesting mutagenesis independent of UV radiation [225,247]. The contrast between this observation and the association of *NRAS* mutation with skin with CSD suggests that detecting the UV signature mutations can help diagnose melanoma.

In melanomas bearing *NRAS* mutations, targeted therapy has shown limited effectiveness and thus immune therapies and chemotherapy are generally used. Direct targeting of N-ras is difficult but most research has focused on inhibiting the farnesylation and subsequent activation of N-ras [248,249]. Subsequently, farnesyl transferase inhibitors (FTIs), such as lonafarnib and tipifarnib, were developed and shown to induce apoptosis in mouse and human melanoma cells [42,43]. However, clinical trials have failed to show efficacy for FTIs, as N-ras becomes activated through alternative post-translational modifications [249,250,251].

Besides N-ras, ongoing research has focused on targeting its downstream signals. MEK inhibitors have been identified as potential therapy for *NRAS* mutants. In particular, binimetinib has been shown in phase 3 trials to have improved survival and response rate compared to dacarbazine in *NRAS*-mutant melanoma [252]. Other clinical trials in progress are studying the potential of combining MEK inhibitors with PI3K, RAF, and cell cycle inhibitors [249,253,254,255,256]. Overall, the options to treat *NRAS*-mutated melanomas remain relatively scarce despite ongoing trials with signaling pathway targets.

#### 3.3.3. C-KIT

C-KIT is a receptor tyrosine kinase (RTK) directly responsible for binding to growth factors as the first signal down the MAPK and PI3K pathways [15,44,85,257]. The majority of *c-KIT* mutations are found in mucosal and acral melanomas, as well as in melanomas arising from skin with CSD [44,258]. Presence of these mutations has also been associated with worse survival as compared with wild-type melanomas [259].

Because of the relative rarity of *c-KIT* mutations, the understanding of targeted therapy to treat melanoma is scarce. Imatinib and nilotinib are the main c-KIT inhibitors that have been studied in melanoma. In a pair of phase 2 trials, imatinib was shown to have significant clinical response in tumors bearing *c-KIT* mutations as compared with wild-type tumors [45,260]. Nilotinib has similarly shown promise in the treatment of patients with *c-KIT* mutations in phase 2 trials [46,47].

#### 3.3.4. Immune Checkpoint (CTLA-4, PD-1, PD-L1)

Immune checkpoint inhibitors represent a novel class of drugs that have increasingly been used in melanoma therapy. By targeting the inactivation signals of the immune system, immunotherapy acts to stimulate the intrinsic immune response to cancer. The main targeted signals are cytotoxic T-lymphocyte antigen-4 (CTLA-4) and programmed death-1 (PD-1). Normally, antigen-presenting cells (APCs) co-stimulate T cells through both binding of the antigen-major histocompatibility complex (MHC) to the T cell receptor (TCR) and of CD80/86 to the CD28 receptor on T cells. As T cells become activated, CTLA-4 is upregulated and binds to the CD28 receptor with greater affinity than does CD80/86. Through this negative feedback, the T cell response is downregulated [66,261]. In peripheral tissue, binding of PD-1 on T cells to PD-L1 on normal or tumor cells reduces the activity and proliferation of T cells [262,263]. Monotherapy and combined therapy with CTLA-4 antibodies (ipilimumab) and PD-1 antibodies (pembrolizumab, nivolumab) have been shown to elicit a positive response against melanoma [67,68,69,264].

While immunotherapy shows promising potential, predicting therapeutic response has proven difficult. PD-L1 has been studied as a potential marker to predict response to immunotherapy without much avail. PD-L1 is estimated to be overexpressed in 45% of melanoma tumor samples [265]. In these samples, PD-L1 expression was associated with improved response rates to checkpoint therapy; however, absence of expression did not rule out the rate or degree of response to combined or monotherapy [265]. Other studies have shown similar patterns of response depending on positive PD-L1 expression [266,267]. Recently, a pair of studies has shown that melanoma cells release vesicles with PD-L1 on the surface and that such exosomal PD-L1 levels correlates with response to anti-PD-1 therapy [268,269]. Thus, identification of PD-L1 expression in tumors may serve as an effective first step in determining treatment with checkpoint therapy serving as a first-line option in tumors with positive expression and as reserve treatment in tumors with negative expression [270].

The expression of PD-1 and CTLA-4 on melanoma cells is not well studied. PD-1 has generally been studied for its expression on immune cells, such as T cells, but receptors have been identified on melanoma cells as well. In fact, the PD-1 receptor was shown to promote growth of melanoma and, as such, bears significance for choosing proper therapy [271]. Similarly, CTLA-4 has been found to be expressed on melanoma cells and to drive tumor formation [272]. An Italian study also found that serum CTLA-4 may serve as a novel biomarker in predicting favorable response to ipilimumab [273]. Timely identification of these markers is crucial for determining treatment for melanoma patients. 

## 4. Summary and Conclusions

The incidence of melanoma continues to rapidly increase in the United States, representing a serious public health problem. Early detection of both cutaneous and uveal melanoma represents one of the most crucial ways of reducing the clinical and financial burden of the disease [274]. Understanding the molecular pathology of melanoma marks a significant stride towards that end. Detecting molecular markers aids diagnosis when histological assessment is challenging. Identification of common mutations, such as *BRAF* or *NRAS*, not only helps to narrow down diagnosis and paint a prognostic picture, but also to guide treatment. The expanding knowledge into vitamin D and melanogenesis represents an untapped reservoir for new clinical therapies of melanoma. Overall, analysis of these markers should become reflexive when melanoma is suspected or after diagnosis to contribute to improving outcomes for patients.

## Figures and Tables

**Figure 1 ijms-21-03535-f001:**
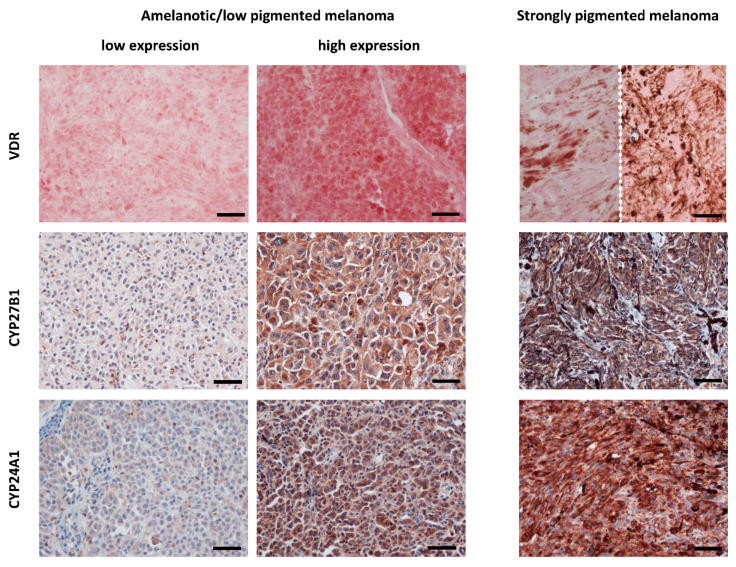
VDR, CYP27B1 and CYP24A1 immunostaining cutaneous melanomas. Left panel presents lack or low VDR, CYP27B1 and CYP24A1 expression in amelanotic/low pigmented human melanomas obtained from tissues of patients treated in Oncology Center, Bydgoszcz, Poland. Middle panel presents high expression of VDR, CYP27B1 and CYP24A1 expression in amelanotic/low pigmented melanomas. Right panel presents VDR, CYP27B1 and CYP24A1 expression in strongly pigmented melanomas (images of VDR from two different cases are separated with dotted line). VDR was labelled with rat antibody (clone 9A7; Abcam, Cambridge, MA, USA; a dilution 1:75) and visualized with Red AP Substrate (Vector Laboratories, Burlingame, CA, USA). CYP27B1 and CYP24A1 were labelled with rabbit antibody (clone H-90, Santa Cruz Biotechnology, Santa Cruz, CA, USA, a dilution of 1:75) and mouse antibody (Abcam, Cambridge, UK, dilution 1:40), respectively, followed by visualization with ImmPACT NovaRED substrate (Vector Laboratories, Burlingame, CA, USA) and counterstaining with hematoxylin [23,143,144]. Scale bars: 50 μm.

**Table 1 ijms-21-03535-t001:** Summary of melanoma mutations.

Gene	Incidence	Hallmarks of Tumors
**Cutaneous Melanoma**		
*BRAF*	40–60%	Superficial spreading subtype; younger patients; non-CSD skin [14,15,16]
*NRAS*	15–30%	Nodular subtype; CSD skin [17,18]
*KIT*	1–2%	Mucosal and acral types; CSD skin [16,19]
*CDKN2A*	25–40% (familial)	Superficial spreading subtype; dysplastic nevus syndrome [20,21,22]
*VDR*	Unknown	Inverse correlation with tumor progression and mitotic rates [23,24]
*MC1R*	Variants in up to 60%	Fair skin, red hair phenotype; presentation on arms [25,26]
*MITF*	1–2% (familial)	Direct correlation with survival and negative lymph node status [27,28]
**Uveal Melanoma**		
*GNAQ/GNA11*	80–90%	Present in most cases of uveal melanoma; rarely cutaneous melanoma; benign blue and uveal nevi [29]
*BAP1*	8–50% (familial)	High metastatic risk; BAPoma (atypical spitzoid tumor) [30,31,32]
*SF3B1*	10–21%	Intermediate metastatic risk; younger patients [33,34,35,36]
*EIF1AX*	13–21%	Low metastatic risk; good prognosis [33,34,36,37]

**Table 2 ijms-21-03535-t002:** Function of and known clinical therapies targeting melanoma mutations.

Marker	Activity	Targeted Therapy
**Cutaneous Melanoma**		
B-raf	Protein kinase along the MAPK pathway; most common mutation	Vemurafenib, dabrafenib, encorafenib [14,38,39,40]
N-ras	GTPase with signal transduction along the MAPK and PI3K pathways	Phase II trials of FTIs, lonafarnib and tipifarnib (NCT00060125 and NCT00281957) [15,41,42,43]
c-Kit	Growth factor-binding RTK; first signal along the MAPK and PI3K pathways	Phase II trials of imatinib and nilotinib; phase II trial of regorafenib (NCT02501551) [15,44,45,46,47]
*CDKN2A*	Encodes p16 and p14ARF to regulate cell cycle and apoptosis	Phase II trial of CDK inhibitor, flavopiridol (NCT00005971) [48,49,50,51]
VDR	Binds active vitamin D to mediate various downstream functions	Phase II trial of high-dose vitamin D (ACTRN12609000351213); phase III trial of vitamin D supplementation (NCT01748448) [52,53,54,55,56]
MC1R	Binds MSH and ACTH to regulate melanogenesis and skin pigmentation	None [25]
MITF	Regulates melanocyte development, differentiation, and function	None [57,58,59]
Melanin	Pigment that scavenges free radicals	None [60,61,62]
TYR/TRP1/TRP2	Proteins related to melanin synthesis	None [63,64]
HAPLN1	ECM component associated with age-related loss	None [65]
CTLA4/PD-1/PD-L1	Downregulates the T cell immune response	Ipilimumab, pembrolizumab, and nivolumab [66,67,68,69]
**Uveal Melanoma**		
GNAQ/GNA11	G protein alpha subunits involved in the MAPK and PI3K pathways	None [70,71,72]
BAP1	Deubiquitinase involved in cell cycle progression	Phase II trial of PARP inhibitor, niraparib (NCT03207347) [73,74]
SF3B1	Splicing factor subunit	None [34,75]
EIF1AX	Eukaryotic translation initiation factor that stabilizes ribosome	None [34,76]

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
