# Peer review of "Current Molecular Markers of Melanoma and Treatment Targets"

_ijms, 2020, doi:10.3390/ijms21103535_

Round 1

Reviewer 1 Report

Authors performed a really good and complete review of the most common mutations usually observed in cutaneous and uveal melanoma. Only minor revisions are needed to improve the quality of the paper. I suggest to substitute the 2006 WHO classification of cutaneous melanoma into four major subtypes: superficial spreading, nodular, lentigo maligna, and acral lentiginous with the recent 2018 WHO classification of melanocytic lesions considering sun exposure. Better quality histological figures could be reported.

Author Response

We greatly appreciate the reviewer effort to improve our presentations, point-by-point response is below.

  1. I suggest to substitute the 2006 WHO classification of cutaneous melanoma into four major subtypes: superficial spreading, nodular, lentigo maligna, and acral lentiginous with the recent 2018 WHO classification of melanocytic lesions considering sun exposure.

We have rewritten page 2, lines 47-55, to reflect the 2018 WHO classification of melanoma.

  • Better quality histological figures could be reported.

Thank you kindly for your feedback. Due to this feedback, as well as feedback from Reviewer 3, Figure 1 was removed. All references to Figure 1 were also removed accordingly.

Reviewer 2 Report

ijms-789924

I have read the manuscript (ID: ijms-789924) Titled: Molecular Pathology of Melanoma: Diagnosis and Therapeutic Implications, by Authors: Kevin Yang, Allen Oak, Radomir Slominski, Anna Brozyna, Andrzej Slominski, Submitted to section: Molecular Oncology, special Issue Molecular Biology of Melanoma

At the first look, the manuscript is aiming to provide a comprehensive review of the topic of melanoma and various mutations which are connect to this cancer with clinical decision-making implications. However, it is neither comprehensive neither novel.

Lot of issues need to be corrected before the final decision about publication.

Major comments:

  1. Title: manuscript is entitled: Molecular Pathology of Melanoma: Diagnosis and Therapeutic Implications. It obliges to put more insight into molecular pathology
  2. The major problem is that the Authors very often cite review articles, not the original research publications.
  3. Sometimes it is difficult to find sentences in cited publications that refer to the Authors’ statements (detailed information provided in comments).
  4. Several chapters would also benefit on some clarifications, as explained in the following section:

Chapter: 2. Molecular Pathways of Melanoma Formation.

5.       The chapter is well written, but could be more elaborated, it is too general.

  1. Thera are some ambiguities:
  • At the beginning of third paragraph there is a sentence: “In contrast to cutaneous melanoma, uveal melanoma tends to….”. This is the first time in this chapter that implicates that everything what was written above refers to cutaneous melanoma. It should be clarified.
  • In the sentence: “The Hippo pathway has been identified for its role in cell homeostasis and ultimately organ size” it is not clear to which organ it refers? Reference points to the review not original paper. In the cited review there is one reference to the article entitled “The emerging roles of YAP and TAZ in cancer”, there is a reference to the organ size in the abstract and then in one cited publication: Heallen T, et al. Hippo pathway inhibits WNT signaling to restrain cardiomyocyte proliferation and heart size. Science. 2011;332:458–461. – it should be checked and corrected.

Chapter: 3. Molecular markers

7.       This chapter would benefit from some introduction, a few general sentences about biomarkers is recommended

8.       Molecular marker is very unprecise statement, it refers in this manuscript to mutations, antigens or even pigment (melanin) presence. This issue should be refined and those biomarkers should be divided between different categories, as aforementioned or discussed according to to the chosen criterion, for example according to the most promising markers or possible targets for therapy

9.       Only in a few subsections (a, b, c, f and i) treatment strategies based on a given markers are presented, what about others? Are there any therapies based on other markers? If not, it would be worth mentioning

References

10.   To avoid impression of epigonism and producing a genetic paper, it is recommended to change all reviews into original experimental papers. There are 52 review references out of the total of 209 references. It makes difficult to follow original works for a reader and it can also generate some understatements:

  • For example in Chapter 3, subsection j) MC1R, first paragraph, “MC1R responds to melanocyte-stimulating hormone (MSH) to regulate melanogenesis and skin pigmentation. Variants in the MC1R gene have been shown to directly and indirectly induce melanomagenesis. In melanocytes, MC1R activates DNA repair and reduces oxidative stress. Thus, MC1R polymorphisms can exacerbate the UV-induced DNA damage and promote tumor formation [147]. In addition, variants of MC1R upregulate pheomelanin production, which is characterized by a phenotype of fair skin and red hair and susceptibility to UV light [147]”. All this information is probably taken from the review. It is ok to provide general statements from review, but the results from particular experiments should have references to the original papers. This is issue that has to be corrected not only in this case but in whole article. In my opinion subsection i) VDR is the best written and could be an example for the other chapters, with respect to the used references.
  • a) BRAF: Sentence: “Mutations in this oncogene lead to constitutive activation of the MAPK pathway. The most common BRAF mutation is V600E, which represents 80% of alterations in the gene” – lack of reference. Further in this chapter is statement: “Studies have shown that V600E expression is associated with the superficial spreading subtype, younger patient age, and skin sites without chronic sun-induced damage, such as the extremities [27-29] – it should be checked, I couldn’t find information about these associations in this articles.
  • Reference no. 137. Slominski, A.; Semak, I.; Zjawiony, J.; Wortsman, J.; Li, W.; Szczesniewski, A.; Tuckey, R. C.,The Cytochrome P450scc System Opens an Alternate Pathway of Vitamin D3 Metabolism. FEBS J 2005, 272, (16), 4080-90. – should be checked. I couldn’t find information that: “CYP11A1, in addition to adrenals, placenta and sex organs [135], is also expressed in several peripheral tissue including immune system [136, 137]….”
  • I) VDR, in the end of paragraph 4, sentence: “It must also be noted that while there was an inverse correlation between CYP27B1 (enzyme activating vitamin D3) and disease progression [141], such correlation with an inactivating enzyme, CYP24A1, was complex [142].” It should be more specific than statement “complex”.

Figures

  1. Description of Figure 1 should be corrected. Description “Human cutaneous melanoma histological and growth subtypes: A) Radial growth phase. B) Vertical growth phase. C) Nodular subtype. D) Metastatic melanoma” suggests that those are four subtypes of cuntaneous melanoma, when it actually refers to two phases of superficial spreading of melanoma: “early radial growth (Figure 1A), followed by vertical growth (Fig (1B)”. Additionally, one image is presented for nodular melanoma (Figure 1C). It would be beneficial to provide histological images for all types of cutaneous melanoma, clearly distinguish those types and if the phases of growth are presented on the figure, to clearly mark this. Additionally, there is lack of information about the source of the images. This is also true for Figure 2.

Tables

  1. Table 1. and Table 2. – they don’t have references, which should be added

General comment

Bibliography which refers to the review not to the orginal papers

  1. Connolly, K. L.; Nehal, K. S.; Busam, K. J., Lentigo Maligna and Lentigo Maligna Melanoma: Contemporary Issues in Diagnosis and Management. Melanoma Manag 2015, 2, (2), 171-178.
  2. Goydos, J. S.; Shoen, S. L., Acral Lentiginous Melanoma. Cancer Treat Res 2016, 167, 321-329.
  3. Krantz, B. A.; Dave, N.; Komatsubara, K. M.; Marr, B. P.; Carvajal, R. D., Uveal Melanoma: Epidemiology, Etiology, and Treatment of Primary Disease. Clin Ophthalmol 2017, 11, 279-289.
  4. Kaliki, S.; Shields, C. L., Uveal Melanoma: Relatively Rare but Deadly Cancer. Eye (Lond) 2017, 31, (2), 241-257.
  5. Jager, M. J.; Shields, C. L.; Cebulla, C. M.; Abdel-Rahman, M. H.; Grossniklaus, H. E.; Stern, M.-H.; Carvajal, R. D.; Belfort, R. N.; Jia, R.; Shields, J. A.; Damato, B. E., Uveal Melanoma. Nat Rev Dis Primers 2020, 6, (1), 24.
  6. Raman, M.; Chen, W.; Cobb, M. H., Differential Regulation and Properties of Mapks. Oncogene 2007, 26, (22), 3100-12.
  7. Leonardi, G. C.; Falzone, L.; Salemi, R.; Zanghì, A.; Spandidos, D. A.; McCubrey, J. A.; Candido, S.; Libra, M., Cutaneous Melanoma: From Pathogenesis to Therapy (Review). Int J Oncol 2018, 52, (4), 1071-1080.
  8. Burotto, M.; Chiou, V. L.; Lee, J.-M.; Kohn, E. C., The Mapk Pathway across Different Malignancies: A New Perspective. Cancer 2014, 120, (22), 3446-3456.
  9. Yuan, T. L.; Cantley, L. C., Pi3k Pathway Alterations in Cancer: Variations on a Theme. Oncogene 2008, 27, (41), 5497-5510.
  10. Kwong, L. N.; Davies, M. A., Navigating the Therapeutic Complexity of Pi3k Pathway Inhibition in Melanoma. Clin Cancer Res 2013, 19, (19), 5310-9.
  11. Amaro, A.; Gangemi, R.; Piaggio, F.; Angelini, G.; Barisione, G.; Ferrini, S.; Pfeffer, U., The Biology of Uveal Melanoma. Cancer Metastasis Rev 2017, 36, (1), 109-140.
  12. Moroishi, T.; Hansen, C. G.; Guan, K. L., The Emerging Roles of Yap and Taz in Cancer. Nat Rev Cancer 2015, 15, (2), 73-79.
  13. Pfeifer, G. P.; Besaratinia, A., Uv Wavelength-Dependent DNA Damage and Human Non- Melanoma and Melanoma Skin Cancer. Photochem Photobiol Sci 2012, 11, (1), 90-7.
  14. Craig, S.; Earnshaw, C. H.; Virós, A., Ultraviolet Light and Melanoma. J Pathol 2018, 244, (5), 578-585.
  15. Lee, J. J.; Lian, C. G., Molecular Testing for Cutaneous Melanoma: An Update and Review. Arch Pathol Lab Med 2019, 143, (7), 811-820.
  16. . Melis, C.; Rogiers, A.; Bechter, O.; van den Oord, J. J., Molecular Genetic and Immunotherapeutic Targets in Metastatic Melanoma. Virchows Arch 2017, 471, (2), 281-293.
  17. Mancera, N.; Smalley, K. S. M.; Margo, C. E., Melanoma of the Eyelid and Periocular Skin: Histopathologic Classification and Molecular Pathology. Surv Ophthalmol 2019, 64, (3), 272-288.
  18. Gill, M.; Celebi, J. T., B-Raf and Melanocytic Neoplasia. J Am Acad Dermatol 2005, 53, (1),108-14.
  19. Sarkisian, S.; Davar, D., Mek Inhibitors for the Treatment of Nras Mutant Melanoma. Drug Des Devel Ther 2018, 12, 2553-2565.
  20. Amann, V. C.; Ramelyte, E.; Thurneysen, S.; Pitocco, R.; Bentele-Jaberg, N.; Goldinger, S.M.; Dummer, R.; Mangana, J., Developments in Targeted Therapy in Melanoma. Eur J Surg Oncol 2017, 43, (3), 581-593.
  21. Trojaniello, C.; Festino, L.; Vanella, V.; Ascierto, P. A., Encorafenib in Combination with Binimetinib for Unresectable or Metastatic Melanoma with Braf Mutations. Expert Rev Clin Pharmacol 2019, 12, (3), 259-266.
  22. Slipicevic, A.; Herlyn, M., Kit in Melanoma: Many Shades of Gray. J Invest Dermatol 2015, 135, (2), 337-338.
  23. Chattopadhyay, C.; Kim, D. W.; Gombos, D. S.; Oba, J.; Qin, Y.; Williams, M. D.; Esmaeli, B.; Grimm, E. A.; Wargo, J. A.; Woodman, S. E.; Patel, S. P., Uveal Melanoma: From Diagnosis to Treatment and the Science in Between. Cancer 2016, 122, (15), 2299-312.
  24. Soura, E.; Eliades, P. J.; Shannon, K.; Stratigos, A. J.; Tsao, H., Hereditary Melanoma: Update on Syndromes and Management: Genetics of Familial Atypical Multiple Mole Melanoma Syndrome. J Am Acad Dermatol 2016, 74, (3), 395-410.
  25. Vivet-Noguer, R.; Tarin, M.; Roman-Roman, S.; Alsafadi, S., Emerging Therapeutic Opportunities Based on Current Knowledge of Uveal Melanoma Biology. Cancers 2019, 11, (7).
  26. Smit, K. N.; Jager, M. J.; de Klein, A.; Kiliҫ, E., Uveal Melanoma: Towards a Molecular Understanding. Prog Retin Eye Res 2019, 100800-100800
  27. Carlberg, C.; Molnar, F., Current Status of Vitamin D Signaling and Its Therapeutic Applications. Curr Top Med Chem 2012, 12, (6), 528-47.
  28. Bikle, D. D., Vitamin D Metabolism and Function in the Skin. Mol Cell Endocrinol 2011, 347,(1-2), 80-9.
  29. Plum, L. A.; DeLuca, H. F., Vitamin D, Disease and Therapeutic Opportunities. Nat Rev Drug Discov 2010, 9, (12), 941-55.
  30. Pinczewski, J.; Slominski, A., The Potential Role of Vitamin D in the Progression of Benign and Malignant Melanocytic Neoplasms. Exp Dermatol 2010, 19, (10), 860-864.
  31. Slominski, A. T.; Brozyna, A. A.; Skobowiat, C.; Zmijewski, M. A.; Kim, T. K.; Janjetovic, Z.; Oak, A. S.; Jozwicki, W.; Jetten, A. M.; Mason, R. S.; Elmets, C.; Li, W.; Hoffman, R. M.; Tuckey, R. C., On the Role of Classical and Novel Forms of Vitamin D in Melanoma Progression and Management. J Steroid Biochem Mol Biol 2018, 177, 159-170.
  32. Slominski, A. T.; Brożyna, A. A.; Zmijewski, M. A.; Jóźwicki, W.; Jetten, A. 879 M.; Mason, R. S.; Tuckey, R. C.; Elmets, C. A., Vitamin D Signaling and Melanoma: Role of Vitamin D and Its Receptors in Melanoma Progression and Management. Lab Invest 2017, 97, (6), 706-724.
  33. Brożyna, A. A.; Hoffman, R. M.; Slominski, A. T., Relevance of Vitamin D in Melanoma Development, Progression and Therapy. Anticancer Res 2020, 40, (1), 473-489.
  34. Tuckey, R. C.; Cheng, C. Y. S.; Slominski, A. T., The Serum Vitamin D Metabolome: What We Know and What Is Still to Discover. J Steroid Biochem Mol Biol 2019, 186, 4-21.
  35. Slominski, A. T.; Manna, P. R.; Tuckey, R. C., On the Role of Skin in the Regulation of Local and Systemic Steroidogenic Activities. Steroids 2015, 103, 72-88.
  36. Feller, L.; Khammissa, R. A. G.; Kramer, B.; Altini, M.; Lemmer, J., Basal Cell Carcinoma, Squamous Cell Carcinoma and Melanoma of the Head and Face. Head Face Med 2016, 12, 11-11.
  37. Weinstein, D.; Leininger, J.; Hamby, C.; Safai, B., Diagnostic and Prognostic Biomarkers in Melanoma. J Clin Aesthet Dermatol 2014, 7, (6), 13-24.
  38. Levy, C.; Khaled, M.; Fisher, D. E., Mitf: Master Regulator of Melanocyte Development and Melanoma Oncogene. Trends Mol Med 2006, 12, (9), 406-414.
  39. Slominski, A.; Tobin, D. J.; Shibahara, S.; Wortsman, J., Melanin Pigmentation in Mammalian Skin and Its Hormonal Regulation. Physiol Rev 2004, 84, (4), 1155-1228.
  40. Slominski, R. M.; Zmijewski, M. A.; Slominski, A. T., The Role of Melanin Pigment in Melanoma. Exp Dermatol 2015, 24, (4), 258-259.
  41. Slominski, A. T.; Zmijewski, M. A.; Plonka, P. M.; Szaflarski, J. P.; Paus, R., How Uv Light Touches the Brain and Endocrine System through Skin, and Why. Endocrinology 2018, 159,(5), 1992-2007.
  42. Zitvogel, L.; Kroemer, G., Targeting Pd-1/Pd-L1 Interactions for Cancer Immunotherapy. Oncoimmunology 2012, 1, (8), 1223-1225.
  43. Postow, M. A.; Callahan, M. K.; Wolchok, J. D., Immune Checkpoint Blockade in Cancer Therapy. J Clin Oncol 2015, 33, (17), 1974-1982.
  44. Mahoney, K. M.; Freeman, G. J.; McDermott, D. F., The Next Immune-Checkpoint Inhibitors: Pd-1/Pd-L1 Blockade in Melanoma. Clin Ther 2015, 37, (4), 764-782.
  45. Patel, S. P.; Kurzrock, R., Pd-L1 Expression as a Predictive Biomarker in Cancer Immunotherapy. Mol Cancer Ther 2015, 14, (4), 847-856.
  46. Sunshine, J.; Taube, J. M., Pd-1/Pd-L1 Inhibitors. Curr Opin Pharmacol 2015, 23, 32-38.
  47. Lipson, E. J.; Forde, P. M.; Hammers, H.-J.; Emens, L. A.; Taube, J. M.; Topalian, S. L., Antagonists of Pd-1 and Pd-L1 in Cancer Treatment. Semin Oncol 2015, 42, (4), 587-600.
  48. Topalian, S. L.; Taube, J. M.; Anders, R. A.; Pardoll, D. M., Mechanism-Driven Biomarkers to Guide Immune Checkpoint Blockade in Cancer Therapy. Nat Rev Cancer 2016, 16, (5), 275-287.
  1. Slominski, A.; Paus, R.; Mihm, M. C., Inhibition of Melanogenesis as an 1223 Adjuvant Strategy in the Treatment of Melanotic Melanomas: Selective Review and Hypothesis. Anticancer Res 1998, 18, (5B), 3709-15.

Author Response

We greatly appreciate the reviewer comments. The manuscript was corrected following the critique, and the point-by-point response is listed below.

Reviewer #2

I have read the manuscript (ID: ijms-789924) Titled: Molecular Pathology of Melanoma: Diagnosis and Therapeutic Implications, by Authors: Kevin Yang, Allen Oak, Radomir Slominski, Anna Brozyna, Andrzej Slominski, Submitted to section: Molecular Oncology, special Issue Molecular Biology of Melanoma

At the first look, the manuscript is aiming to provide a comprehensive review of the topic of melanoma and various mutations which are connect to this cancer with clinical decision-making implications. However, it is neither comprehensive neither novel.

Lot of issues need to be corrected before the final decision about publication.

  • Title: manuscript is entitled: Molecular Pathology of Melanoma: Diagnosis and Therapeutic Implications. It obliges to put more insight into molecular pathology. Thank you kindly for this feedback. We agreed that the main focus of this review is going over diagnostic/prognostic markers as well as current and future treatment modalities. Therefore, the title was changed to “Current Molecular Markers of Melanoma and Treatment Targets”.

Thank you kindly for this feedback. We agreed that the main focus of this review is going over diagnostic/prognostic markers as well as current and future treatment modalities. Therefore, the title was changed to “Current Molecular Markers of Melanoma and Treatment Targets”.

  • The major problem is that the Authors very often cite review articles, not the original research publications.
  • Sometimes it is difficult to find sentences in cited publications that refer to the Authors’ statements (detailed information provided in comments).Thank you kindly for your feedback. These are addressed as described in below comments.

Thank you kindly for your feedback. These are addressed as described in below comments.

  • Several chapters would also benefit on some clarifications, as explained in the following section: Chapter: 2. Molecular Pathways of Melanoma Formation. The chapter is well written, but could be more elaborated, it is too general.We have elaborated in this section to discuss more of the downstream effectors of the MAPK and PI3K pathways.

We have elaborated in this section to discuss more of the downstream effectors of the MAPK and PI3K pathways.

  • Thera are some ambiguities:

At the beginning of third paragraph there is a sentence: “In contrast to cutaneous melanoma, uveal melanoma tends to….”. This is the first time in this chapter that implicates that everything what was written above refers to cutaneous melanoma. It should be clarified.

We have changed “Melanomagenesis” to “Cutaneous melanomagenesis” in line 141 to clarify this ambiguity.

In the sentence: “The Hippo pathway has been identified for its role in cell homeostasis and ultimately organ size” it is not clear to which organ it refers? Reference points to the review not original paper. In the cited review there is one reference to the article entitled “The emerging roles of YAP and TAZ in cancer”, there is a reference to the organ size in the abstract and then in one cited publication: Heallen T, et al. Hippo pathway inhibits WNT signaling to restrain cardiomyocyte proliferation and heart size. Science. 2011;332:458–461. – it should be checked and corrected.

We have added references in lines 160-161 to reflect the role of Hippo in controlling the size of multiple mammalian organs, including heart, liver, pancreas, and intestine.

Chapter: 3. Molecular markers

  • This chapter would benefit from some introduction, a few general sentences about biomarkers is recommended.
  • Molecular marker is very unprecise statement, it refers in this manuscript to mutations, antigens or even pigment (melanin) presence. This issue should be refined and those biomarkers should be divided between different categories, as aforementioned or discussed according to to the chosen criterion, for example according to the most promising markers or possible targets for therapy.

We have divided the markers according to their most appropriate characteristic into three primary sections: prognostic/diagnostic markers, members of the melanin synthesis pathway, and therapeutic targets. Discussion of these was added further in the introduction of the overall section.

  • Only in a few subsections (a, b, c, f and i) treatment strategies based on a given markers are presented, what about others? Are there any therapies based on other markers? If not, it would be worth mentioning

For those subsections without treatment strategies, Table 2 reflects an absence of current targeted therapy being investigated clinically.

References

  • To avoid impression of epigonism and producing a genetic paper, it is recommended to change all reviews into original experimental papers. There are 52 review references out of the total of 209 references. It makes difficult to follow original works for a reader and it can also generate some understatements:

For example in Chapter 3, subsection j) MC1R, first paragraph, “MC1R responds to melanocyte-stimulating hormone (MSH) to regulate melanogenesis and skin pigmentation. Variants in the MC1R gene have been shown to directly and indirectly induce melanomagenesis. In melanocytes, MC1R activates DNA repair and reduces oxidative stress. Thus, MC1R polymorphisms can exacerbate the UV-induced DNA damage and promote tumor formation [147]. In addition, variants of MC1R upregulate pheomelanin production, which is characterized by a phenotype of fair skin and red hair and susceptibility to UV light [147]”. All this information is probably taken from the review. It is ok to provide general statements from review, but the results from particular experiments should have references to the original papers. This is issue that has to be corrected not only in this case but in whole article. In my opinion subsection i) VDR is the best written and could be an example for the other chapters, with respect to the used references.

We have added references to the original works in these lines as suggested. However, we also believe that citing high-impact or unique reviews are preferable at times in this review since many different aspects of melanoma are addressed. If all the original publications were cited, there would be more than 1,000 citations. If the reviewer has specific areas in which additional original citations are needed, we will include them as well.

  • BRAF: Sentence: “Mutations in this oncogene lead to constitutive activation of the MAPK pathway. The most common BRAF mutation is V600E, which represents 80% of alterations in the gene” – lack of reference.

We have included the reference for this statement in line 525.

Further in this chapter is statement: “Studies have shown that V600E expression is associated with the superficial spreading subtype, younger patient age, and skin sites without chronic sun-induced damage, such as the extremities [27-29] – it should be checked, I couldn’t find information about these associations in this articles.

As suggested, the references were checked and the information in these statements was indeed presented and appropriately referenced from these articles.

Reference no. 137. Slominski, A.; Semak, I.; Zjawiony, J.; Wortsman, J.; Li, W.; Szczesniewski, A.; Tuckey, R. C.,The Cytochrome P450scc System Opens an Alternate Pathway of Vitamin D3 Metabolism. FEBS J 2005, 272, (16), 4080-90. – should be checked. I couldn’t find information that: “CYP11A1, in addition to adrenals, placenta and sex organs [135], is also expressed in several peripheral tissue including immune system [136, 137]….”

As suggested, the references were checked with reference 136 containing information about expression in immune system and reference 137 containing information about expression in adrenals.

  1. I) VDR, in the end of paragraph 4, sentence: “It must also be noted that while there was an inverse correlation between CYP27B1 (enzyme activating vitamin D3) and disease progression [141], such correlation with an inactivating enzyme, CYP24A1, was complex [142].”It should be more specific than statement “complex”.

We have elaborated on this statement to reflect the unexpected correlation and possible mechanism in lines 408-410.

Figures

  • Description of Figure 1 should be corrected. Description “Human cutaneous melanoma histological and growth subtypes: A) Radial growth phase. B) Vertical growth phase. C) Nodular subtype. D) Metastatic melanoma” suggests that those are four subtypes of cuntaneous melanoma, when it actually refers to two phases of superficial spreading of melanoma: “early radial growth (Figure 1A), followed by vertical growth (Fig (1B)”. Additionally, one image is presented for nodular melanoma (Figure 1C). It would be beneficial to provide histological images for all types of cutaneous melanoma, clearly distinguish those types and if the phases of growth are presented on the figure, to clearly mark this. Additionally, there is lack of information about the source of the images. This is also true for Figure 2.

Due to this feedback, as well as feedback from Reviewer 1, Figure 1 was removed. All references to Figure 1 were also removed accordingly. More information has been provided in the figure legend for Figure 2. Specifically, these images were taken from former research with references now included in the legend. The images have never been published.

Tables

  • Table 1. and Table 2. – they don’t have references, which should be added

We have added the references to the table.

General comment

  • Bibliography which refers to the review not to the orginal papers
  • Connolly, K. L.; Nehal, K. S.; Busam, K. J., Lentigo Maligna and Lentigo Maligna Melanoma: Contemporary Issues in Diagnosis and Management. Melanoma Manag 2015, 2, (2), 171-178.
  • Goydos, J. S.; Shoen, S. L., Acral Lentiginous Melanoma. Cancer Treat Res 2016, 167, 321-329.
  • Krantz, B. A.; Dave, N.; Komatsubara, K. M.; Marr, B. P.; Carvajal, R. D., Uveal Melanoma: Epidemiology, Etiology, and Treatment of Primary Disease. Clin Ophthalmol 2017, 11, 279-289.
  • Kaliki, S.; Shields, C. L., Uveal Melanoma: Relatively Rare but Deadly Cancer. Eye (Lond) 2017, 31, (2), 241-257.
  • Jager, M. J.; Shields, C. L.; Cebulla, C. M.; Abdel-Rahman, M. H.; Grossniklaus, H. E.; Stern, M.-H.; Carvajal, R. D.; Belfort, R. N.; Jia, R.; Shields, J. A.; Damato, B. E., Uveal Melanoma. Nat Rev Dis Primers 2020, 6, (1), 24.
  • Raman, M.; Chen, W.; Cobb, M. H., Differential Regulation and Properties of Mapks. Oncogene 2007, 26, (22), 3100-12.
  • Leonardi, G. C.; Falzone, L.; Salemi, R.; Zanghì, A.; Spandidos, D. A.; McCubrey, J. A.; Candido, S.; Libra, M., Cutaneous Melanoma: From Pathogenesis to Therapy (Review). Int J Oncol 2018, 52, (4), 1071-1080.
  • Burotto, M.; Chiou, V. L.; Lee, J.-M.; Kohn, E. C., The Mapk Pathway across Different Malignancies: A New Perspective. Cancer 2014, 120, (22), 3446-3456.
  • Yuan, T. L.; Cantley, L. C., Pi3k Pathway Alterations in Cancer: Variations on a Theme. Oncogene 2008, 27, (41), 5497-5510.
  • Kwong, L. N.; Davies, M. A., Navigating the Therapeutic Complexity of Pi3k Pathway Inhibition in Melanoma. Clin Cancer Res 2013, 19, (19), 5310-9.
  • Amaro, A.; Gangemi, R.; Piaggio, F.; Angelini, G.; Barisione, G.; Ferrini, S.; Pfeffer, U., The Biology of Uveal Melanoma. Cancer Metastasis Rev 2017, 36, (1), 109-140.
  • Moroishi, T.; Hansen, C. G.; Guan, K. L., The Emerging Roles of Yap and Taz in Cancer. Nat Rev Cancer 2015, 15, (2), 73-79.
  • Pfeifer, G. P.; Besaratinia, A., Uv Wavelength-Dependent DNA Damage and Human Non- Melanoma and Melanoma Skin Cancer. Photochem Photobiol Sci 2012, 11, (1), 90-7.
  • Craig, S.; Earnshaw, C. H.; Virós, A., Ultraviolet Light and Melanoma. J Pathol 2018, 244, (5), 578-585.
  • Lee, J. J.; Lian, C. G., Molecular Testing for Cutaneous Melanoma: An Update and Review. Arch Pathol Lab Med 2019, 143, (7), 811-820.
  • . Melis, C.; Rogiers, A.; Bechter, O.; van den Oord, J. J., Molecular Genetic and Immunotherapeutic Targets in Metastatic Melanoma. Virchows Arch 2017, 471, (2), 281-293.
  • Mancera, N.; Smalley, K. S. M.; Margo, C. E., Melanoma of the Eyelid and Periocular Skin: Histopathologic Classification and Molecular Pathology. Surv Ophthalmol 2019, 64, (3), 272-288.
  • Gill, M.; Celebi, J. T., B-Raf and Melanocytic Neoplasia. J Am Acad Dermatol 2005, 53, (1),108-14.
  • Sarkisian, S.; Davar, D., Mek Inhibitors for the Treatment of Nras Mutant Melanoma. Drug Des Devel Ther 2018, 12, 2553-2565.
  • Amann, V. C.; Ramelyte, E.; Thurneysen, S.; Pitocco, R.; Bentele-Jaberg, N.; Goldinger, S.M.; Dummer, R.; Mangana, J., Developments in Targeted Therapy in Melanoma. Eur J Surg Oncol 2017, 43, (3), 581-593.
  • Trojaniello, C.; Festino, L.; Vanella, V.; Ascierto, P. A., Encorafenib in Combination with Binimetinib for Unresectable or Metastatic Melanoma with Braf Mutations. Expert Rev Clin Pharmacol 2019, 12, (3), 259-266.
  • Slipicevic, A.; Herlyn, M., Kit in Melanoma: Many Shades of Gray. J Invest Dermatol 2015, 135, (2), 337-338.
  • Chattopadhyay, C.; Kim, D. W.; Gombos, D. S.; Oba, J.; Qin, Y.; Williams, M. D.; Esmaeli, B.; Grimm, E. A.; Wargo, J. A.; Woodman, S. E.; Patel, S. P., Uveal Melanoma: From Diagnosis to Treatment and the Science in Between. Cancer 2016, 122, (15), 2299-312.
  • Soura, E.; Eliades, P. J.; Shannon, K.; Stratigos, A. J.; Tsao, H., Hereditary Melanoma: Update on Syndromes and Management: Genetics of Familial Atypical Multiple Mole Melanoma Syndrome. J Am Acad Dermatol 2016, 74, (3), 395-410.
  • Vivet-Noguer, R.; Tarin, M.; Roman-Roman, S.; Alsafadi, S., Emerging Therapeutic Opportunities Based on Current Knowledge of Uveal Melanoma Biology. Cancers 2019, 11, (7).
  • Smit, K. N.; Jager, M. J.; de Klein, A.; Kiliҫ, E., Uveal Melanoma: Towards a Molecular Understanding. Prog Retin Eye Res 2019, 100800-100800
  • Carlberg, C.; Molnar, F., Current Status of Vitamin D Signaling and Its Therapeutic Applications. Curr Top Med Chem 2012, 12, (6), 528-47.
  • Bikle, D. D., Vitamin D Metabolism and Function in the Skin. Mol Cell Endocrinol 2011, 347,(1-2), 80-9.
  • Plum, L. A.; DeLuca, H. F., Vitamin D, Disease and Therapeutic Opportunities. Nat Rev Drug Discov 2010, 9, (12), 941-55.
  • Pinczewski, J.; Slominski, A., The Potential Role of Vitamin D in the Progression of Benign and Malignant Melanocytic Neoplasms. Exp Dermatol 2010, 19, (10), 860-864.
  • Slominski, A. T.; Brozyna, A. A.; Skobowiat, C.; Zmijewski, M. A.; Kim, T. K.; Janjetovic, Z.; Oak, A. S.; Jozwicki, W.; Jetten, A. M.; Mason, R. S.; Elmets, C.; Li, W.; Hoffman, R. M.; Tuckey, R. C., On the Role of Classical and Novel Forms of Vitamin D in Melanoma Progression and Management. J Steroid Biochem Mol Biol 2018, 177, 159-170.
  • Slominski, A. T.; Brożyna, A. A.; Zmijewski, M. A.; Jóźwicki, W.; Jetten, A. 879 M.; Mason, R. S.; Tuckey, R. C.; Elmets, C. A., Vitamin D Signaling and Melanoma: Role of Vitamin D and Its Receptors in Melanoma Progression and Management. Lab Invest 2017, 97, (6), 706-724.
  • Brożyna, A. A.; Hoffman, R. M.; Slominski, A. T., Relevance of Vitamin D in Melanoma Development, Progression and Therapy. Anticancer Res 2020, 40, (1), 473-489.
  • Tuckey, R. C.; Cheng, C. Y. S.; Slominski, A. T., The Serum Vitamin D Metabolome: What We Know and What Is Still to Discover. J Steroid Biochem Mol Biol 2019, 186, 4-21.
  • Slominski, A. T.; Manna, P. R.; Tuckey, R. C., On the Role of Skin in the Regulation of Local and Systemic Steroidogenic Activities. Steroids 2015, 103, 72-88.
  • Feller, L.; Khammissa, R. A. G.; Kramer, B.; Altini, M.; Lemmer, J., Basal Cell Carcinoma, Squamous Cell Carcinoma and Melanoma of the Head and Face. Head Face Med 2016, 12, 11-11.
  • Weinstein, D.; Leininger, J.; Hamby, C.; Safai, B., Diagnostic and Prognostic Biomarkers in Melanoma. J Clin Aesthet Dermatol 2014, 7, (6), 13-24.
  • Levy, C.; Khaled, M.; Fisher, D. E., Mitf: Master Regulator of Melanocyte Development and Melanoma Oncogene. Trends Mol Med 2006, 12, (9), 406-414.
  • Slominski, A.; Tobin, D. J.; Shibahara, S.; Wortsman, J., Melanin Pigmentation in Mammalian Skin and Its Hormonal Regulation. Physiol Rev 2004, 84, (4), 1155-1228.
  • Slominski, R. M.; Zmijewski, M. A.; Slominski, A. T., The Role of Melanin Pigment in Melanoma. Exp Dermatol 2015, 24, (4), 258-259.
  • Slominski, A. T.; Zmijewski, M. A.; Plonka, P. M.; Szaflarski, J. P.; Paus, R., How Uv Light Touches the Brain and Endocrine System through Skin, and Why. Endocrinology 2018, 159,(5), 1992-2007.
  • Zitvogel, L.; Kroemer, G., Targeting Pd-1/Pd-L1 Interactions for Cancer Immunotherapy. Oncoimmunology 2012, 1, (8), 1223-1225.
  • Postow, M. A.; Callahan, M. K.; Wolchok, J. D., Immune Checkpoint Blockade in Cancer Therapy. J Clin Oncol 2015, 33, (17), 1974-1982.
  • Mahoney, K. M.; Freeman, G. J.; McDermott, D. F., The Next Immune-Checkpoint Inhibitors: Pd-1/Pd-L1 Blockade in Melanoma. Clin Ther 2015, 37, (4), 764-782.
  • Patel, S. P.; Kurzrock, R., Pd-L1 Expression as a Predictive Biomarker in Cancer Immunotherapy. Mol Cancer Ther 2015, 14, (4), 847-856.
  • Sunshine, J.; Taube, J. M., Pd-1/Pd-L1 Inhibitors. Curr Opin Pharmacol 2015, 23, 32-38.
  • Lipson, E. J.; Forde, P. M.; Hammers, H.-J.; Emens, L. A.; Taube, J. M.; Topalian, S. L., Antagonists of Pd-1 and Pd-L1 in Cancer Treatment. Semin Oncol 2015, 42, (4), 587-600.
  • Topalian, S. L.; Taube, J. M.; Anders, R. A.; Pardoll, D. M., Mechanism-Driven Biomarkers to Guide Immune Checkpoint Blockade in Cancer Therapy. Nat Rev Cancer 2016, 16, (5), 275-287.
  • Slominski, A.; Paus, R.; Mihm, M. C., Inhibition of Melanogenesis as an 1223 Adjuvant Strategy in the Treatment of Melanotic Melanomas: Selective Review and Hypothesis. Anticancer Res 1998, 18, (5B), 3709-15. 
  1. In addition to the references listed above, we have added references to original papers when appropriate. We also believe that citations of reviews are highly appropriate in this case (see above). We have added the following sentence into the acknowledgement “In compliance with journal regulations, many original papers on the topic are cited in the referenced reviews.”.

Reviewer 3 Report

The paper by Yang et al. entitled "Molecular Pathology of Melanoma: Diagnosis and Therapeutic Implications" are a review paper on the current updated knowledge of melanoma management. Overall the paper is well written and structured and comprehensively summarizes papers about the above issue.

I would like to raise the following minor concern.

1. Figure 1 can be improved. The pictures fail to appropriately represent the each growth subtype. Fig.1 may not be necessary to this review. 

Author Response

  • Figure 1 can be improved. The pictures fail to appropriately represent the each growth subtype. Fig.1 may not be necessary to this review. 

Thank you kindly for your feedback. Due to this feedback, as well as feedback from Reviewer 1, Figure 1 was removed. All references to Figure 1 were also removed accordingly.

Round 2

Reviewer 2 Report

I have 3 notes in total

  1. I cannot know if Figure 1 is in the final version (the reference in the text is removed but the picture is still visible)
  2. There are still no sources of the histopathology pictures
  3. The authors need correct the numbering of the drawings.

Author Response

We thank the reviewer for the attention to the detail. The manuscript has been corrected as requested

  1. The figure was removed. It was a "ghost" figure on the version with track changes.
  2. The source has been provided in the figure legend
  3. The numbering was corrected, where appropriate
  4. The changes are marked in re